# The Effect of Low Intensity Resistance Training with Blood Flow Restriction on Fall Resistance in Middle-Aged and Older Adults: A Meta-Analysis

**DOI:** 10.3390/ijerph20064723

**Published:** 2023-03-07

**Authors:** Shufan Li, Peng Wang, Xin Xin, Xiaojing Zhou, Jing Wang, Jinlei Zhao, Xing Wang

**Affiliations:** 1School of Physical Education, Shanghai University of Sport, Shanghai 200438, China; 2Department of Sports and Health, Shanghai Lixin University of Accounting and Finance, Shanghai 201620, China

**Keywords:** blood flow restriction, pressurized resistance training, middle-aged and older adults, fall, muscle strength, muscle mass, muscle function, balance, walking, meta-analysis

## Abstract

Objective: To systematically evaluate the effect of low intensity resistance training with blood flow restriction on fall resistance in middle-aged and older adults. Methods: PubMed, The Cochrane Library, Web of Science, Embase, EBSCO host, CNKI Database, VIP, Wanfang Database and CBMdisc were searched, and the retrieval period was from the beginning of each database to 25 July 2022. Randomized controlled trials were collected concerning the intervention of low intensity resistance training with blood flow restriction on lower limb muscle strength, muscle mass, muscle function, balance, walking and other fall resistance indicators in middle-aged and older adults. Cochrane Risk of Bias Tool was used for methodological quality assessment of the included literature. Statistical analysis was performed using RevMan 5.4 software and Stata 15.1. Results: A total of 14 randomized controlled trials (419 participants in total) were included in the study. Meta-analysis results revealed that low intensity resistance training with blood flow restriction significantly improved lower limb muscle strength (SMD = 0.51, 95%CI: [0.28, 0.74], *p* < 0.0001), lower limb muscle mass (MD = 1.99, 95%CI: [0.77, 3.22], *p* = 0.001) and walking ability (SMD = −0.89, 95% CI. [−1.71, −0.06], *p* = 0.03), while there was no apparent intervention effect on lower limb muscle function (SMD = 0.25, 95%CI: [−0.23, 0.73], *p* = 0.31) and balance (SMD = 0.22, 95%CI: [−0.08, 0.52], *p* = 0.15). The results of subgroup analysis showed that the intervention effect of low intensity resistance training with blood flow restriction on lower limb muscle strength was more significant in subjects aged 55–64 years, with exercise cycles of 4–8 weeks, exercise frequency of three times per week, exercise intensity of 20–30% 1RM, and vascular flow blocking pressure ≥ 120 mmHg. Conclusion: Low intensity resistance training with blood flow restriction can effectively improve lower limb muscle strength, muscle mass and walking ability in middle-aged and older adults, and can serve as an important form of fall resistance training for the older adults.

## 1. Introduction

According to the World Health Organization (WHO), the number and proportion of people aged 60 and over is increasing dramatically and is expected to rise to 1.4 billion by 2030 and 2.1 billion by 2050, accounting for 22% of the world’s population [1]. Falls are an essential cause of disability and death in middle-aged and older adults, with about 28–35% of older people falling each year; the incidence of falls rises with age [2]. Decline in muscle strength and muscle mass in middle-aged and older adults is a core trigger for falls [3]. The muscle mass of the human body declines at a rate of 3% to 8% per decade after age 30 [4], and muscle strength declines by 12% to 14% per decade in middle-aged and older adults over age 50 [5]. Changes in muscle strength of lower limbs lead to impact on body movements such as walking, sitting, standing, and climbing ladders in middle-aged and older people, producing a decrease in lower extremity muscle function, balance and gait disorders, which in turn trigger falls [6,7,8]. Therefore, it is of great importance to improve muscle strength, muscle quality and muscle function of lower limbs, maintain and enhance balance and walking ability in middle-aged and older adults for the purpose of reducing the risk of falls.

Blood flow restriction training (BFRT), which originated from KAATSU training, is a training method in which external pressure is applied to the upper and proximal extremities by means of a special compression device to block venous blood flow during exercise [9]. Low intensity resistance training with blood flow restriction (LIRT + BFR) has less stress on the joints and cardiovascular system, making it safer compared to traditional high-intensity resistance training [10], which is suitable for muscle strength interventions in the older population [11]. It has been demonstrated that LIRT + BFR induces fast-twitch muscle fiber to participate in muscle work to maintain muscle strength, increases the secretion of growth hormone and activates the mechanism of muscle growth, effectively increasing muscle strength and mass in middle-aged and older adults [12,13,14]. The increase in muscle mass and strength could improve muscle function [15]. Moreover, LIRT + BFR stimulates angiogenesis and enhances endothelial function, which in turn improves muscle function [16].

A review of previous studies indicated that the increase in muscle strength and muscle mass induced by LIRT + BFR was at least similar to that by conventional HIRT [11]. However, its effects on lower limb muscle function, balance, and walking need to be further explored. Therefore, we addressed the following research questions. Can LIRT + BFR effectively increase muscle strength and muscle mass of lower limbs in middle-aged and older adults? Can it successfully improve lower limb muscle function, balance and walking ability in middle-aged and older adults?

## 2. Research Methods

The study followed the requirements of the Preferred Reporting Items for Systematic Reviews and Meta-Analyses (PRISMA) [17] for the selection and use of research methods. PROSPERO CRD42022379662. 

### 2.1. Literature Search Strategies

Nine databases, including PubMed, The Cochrane Library, Web of Science, Embase, EBSCO host, China National Knowledge Infrastructure (CNKI) database, VIP, Wanfang Database and CBMdisc, were searched by two researchers and supplemented with a literature tracing method. The retrieval period was from the beginning of each database to 25 July 2022. Randomized controlled trials concerning the intervention of low intensity resistance training with blood flow restriction on lower limb muscle strength, muscle mass, muscle function, balance, walking and other fall resistance indicators in middle-aged and older adults were collected. A combination of subject terms and free words was used for the search. English search terms included: blood flow restriction therapy, BFR Therapy, KAATSU, vascular occlusion training, ischemic training, accidental falls, falls, slip and fall, muscle strength, strength, muscle mass, physical functional performance, muscular function, postural balance, balance, walking, middle-aged and older adults, older adults, old people, and randomized controlled trial. Taking Pubmed as an example, the search strategy was: 

#1 Aged [Mesh] OR “middle-aged and older adults” [Title/Abstract] OR “Older adult” [Title/Abstract] OR “Old people” [Title/Abstract].

#2 Blood Flow Restriction Therapy [Mesh] OR “BFR Therapy” [Title/Abstract] OR KAATSU [Title/Abstract] OR “vascular occlusion training” [Title/Abstract] OR “ischemic training” [Title/Abstract].

#3 “accidental falls” [Mesh] OR Falls [Title/Abstract] OR “Slip and Fall” [Title/Abstract] OR “muscle strength” [Title/Abstract] OR “strength” [Title/Abstract] OR muscle mass [Title/Abstract] OR “Physical Functional Performance” [Title/Abstract] OR “muscular function” [Title/Abstract] OR “Postural Balance” [Title/Abstract] OR “balance” [Title/Abstract] OR “Walking” [Title/Abstract] OR “Ambulation” [Title/Abstract]. 

#4 Randomized controlled trial [Publication Type] OR Randomized [Title/Abstract] OR controlled [Title/Abstract] OR trial [Title/Abstract].

#5 #1 AND #2 AND #3 AND #4.

### 2.2. Literature Inclusion and Exclusion Criteria

#### 2.2.1. Inclusion Criteria

The inclusion criteria for the literature were based on the PICOS principles of evidence-based medicine.

(1)Study subjects: middle-aged and older adults between 45 and 89 years of age; no restriction on gender, race, or country; no psychiatric abnormalities or severe perceptual disorders, no musculoskeletal disorders or surgical history, and no major organic diseases.(2)Intervention: Subjects engaged in low intensity resistance training with blood flow restriction (<50% 1RM).(3)Comparison group: subjects performed daily exercise (maintenance of daily life or regular exercise), low-intensity resistance exercise (<50% 1RM), high-intensity resistance exercise (≥70% 1RM), or other exercise. If more than one group of data were compared in the same literature, they were counted as multiple studies.(4)Outcome indicators: the main outcome indicators were muscle strength and muscle mass of the lower limbs. For lower limb muscle strength, 1 or 10 repetitions of maximum strength (1RM/10RM), maximum voluntary contraction (MVC), and isometric moment were selected. For lower limb muscle mass, muscle cross-sectional area (CSA) was selected. The secondary outcome indicators were lower limb muscle function, balance and walking ability. For lower limb muscle function, SPPB (Short Physical Performance Battery) and 30-s sit to stand test were selected. For balance, the balance extension test and single leg stand test (with eyes open and closed) were selected. For walking ability, timed up and go (TUG) and walking time test were selected.(5)Type of research: the included literature was the randomized controlled trial (RCT).

#### 2.2.2. Exclusion Criteria

(1) Non-randomized controlled trials (RCTs); (2) non-English and Chinese literature; (3) animal experiments; (4) incomplete experimental data that could not be calculated (extraction of the data revealed missing data on the mean, standard deviation, and sample size of the outcome indicators measured before and after the intervention in the literature, and the original data were still not available after contacting the authors); (5) full text of the literature was not available; (6) experimental groups were combined intervention studies, such as the combined intervention of low intensity resistance training with blood flow restriction with aerobic exercise and nutritional intake.

### 2.3. Literature Screening and Data Extraction

The literature retrieved from each database was imported into Endnote software for de-duplication (including the same literature retrieved from various databases, duplicate publications). Two researchers (XX, XJZ) screened the literature according to the inclusion and exclusion criteria, respectively, extracted the information and reviewed each other, and in case of disagreement, a third person (JW) decided, jointly, whether to include it. Data extraction included (i) basic information (first author, year of publication, country, gender, age); (ii) experimental characteristics (sample size, baseline characteristics, interventions, exercise intensity, exercise cycle, exercise frequency, vascular flow blocking pressure), and (iii) outcome indicators.

### 2.4. Quality Assessment

A methodological quality assessment of the included literature was performed. In this paper, the risk of bias of the included literature was evaluated in seven aspects according to the risk bias assessment tool recommended in the Cochrane Handbook 5.1.0, which included random sequence generation, allocation concealment, blinding of subjects and study personnel, blinding of outcome assessors, incomplete outcome data, selective reporting, and other biases. For each indicator, “low risk of bias,” “uncertainty of bias,” and “high risk of bias” were used for determination.

### 2.5. Data Processing

RevMan 5.4 software was used for statistical analysis. Heterogeneity tests were performed using *p*-values and I^2^. If there was statistical heterogeneity (I^2^ ≥ 50%, *p* < 0.10) between the results of the studies, a random-effects model was selected, and conversely, a fixed-effects model was selected. Measures with the same measurement instrument were expressed as weighted mean differences (MD) and 95% confidence intervals (CI), otherwise standardized mean differences (SMD) and 95% CI were used. A meta-analysis of all outcome indicators in the included literature was performed using RevMan 5.4, and the Begg test in Stata 15.1 was applied to test for publication bias for outcome indicators for more than 10 studies.

## 3. Results

### 3.1. Literature Search Results

A total of 965 papers were retrieved by computer search, no papers were obtained from other resources supplement, and 876 papers were available after de-duplication. After initial screening by reading the title and abstract, full-text re-screening, and elimination of literature which did not meet the inclusion criteria, 14 papers were finally included in the meta-analysis, as shown in Figure 1.

### 3.2. Basic Characteristics of the Included Literature

Table 1 presents basic information on the included studies, which included 14 RCTs (419 individuals in total) with subjects aged 50–82 years.

### 3.3. Methodological Quality Assessment of the Included Literature

As shown in Figure 2, all 14 papers used randomized controlled trials, of which 13 described the randomization method in detail, three described the allocation concealment scheme, two were blinded to subjects and investigators, seven used blinding of outcome assessors, and the remaining blinding was unclear or not used. Outcome data and the reports were completed for all the literature.

### 3.4. Meta-Analysis Results

#### 3.4.1. Effects of Low Intensity Resistance Training with Blood Flow Restriction on Lower Limb Muscle Strength in Middle-Aged and Older Adults

All fourteen included papers (fifty-one studies in total) compared the differences in group for low intensity resistance training with blood flow restriction with the control group for lower limb muscle strength (Figure 3). The results of the heterogeneity test revealed that I^2^ = 70%, *p* < 0.10, indicating high heterogeneity among studies, so the random effects model was adopted for the analysis. Meta-analysis indicated that the combined effect size SMD = 0.51, 95%CI: (0.28, 0.74), *p* < 0.0001, suggesting a statistically significant difference compared to the control group, and that low intensity resistance training with blood flow restriction was effective in enhancing lower limb muscle strength in middle-aged and older adults.

To explore the heterogeneity caused by different study protocols, the effects of low intensity resistance training with blood flow restriction on lower limb muscle strength were compared with different control groups (daily exercise, LIRT, HIRT and dynamic balance exercise). As shown in Table 2, LIRT + BFR showed higher heterogeneity compared to daily exercise and lower heterogeneity in other groups. LIRT + BFR improved lower limb muscle strength better compared with daily exercise and low-intensity resistance exercise. The effect of high intensity resistance exercise was superior compared to that of LIRT + BFR. There was no significant difference between the two groups comparing LIRT + BFR with dynamic balance exercise.

#### 3.4.2. Effects of Low Intensity Resistance Training with Blood Flow Restriction on Lower Limb Muscle Mass in Middle-Aged and Older Adults

Six of the fourteen included papers (nine studies in total) compared the differences in group for low intensity resistance training with blood flow restriction with the control group for lower limb muscle mass (Figure 4). The results of the heterogeneity test showed that I^2^ = 0%, *p* = 0.77, indicating low heterogeneity among studies. Therefore, a fixed effects model was applied for the analysis. Meta-analysis demonstrated that the combined effect size MD = 1.99, 95%CI: (0.77, 3.22]), *p* = 0.001, indicating a statistically significant difference compared to the control group, and that low intensity resistance training with blood flow restriction effectively improved lower limb muscle mass in middle-aged and older adults.

#### 3.4.3. Effects of Low Intensity Resistance Training with Blood Flow Restriction on Lower Limb Muscle Function in Middle-Aged and Older Adults

Two of the fourteen included papers (three studies in total) compared the differences in group for low intensity resistance training with blood flow restriction with the control group for lower limb muscle function (Figure 5). The results of the heterogeneity test showed that I² = 0%, *p* = 0.50, indicating low heterogeneity among studies, so the fixed effects model was used for analysis. Meta-analysis presented a combined effect size SMD = 0.25, 95%CI: (−0.23, 0.73), *p* = 0.31, with no significant difference between the two groups.

#### 3.4.4. Effects of Low Intensity Resistance Training with Blood Flow Restriction on Balance in Middle-Aged and Older Adults

Two of the fourteen included papers (seven studies in total) compared the differences in group for low intensity resistance training with blood flow restriction with the control group for balance (Figure 6). The results of the heterogeneity test revealed that I^2^ = 44%, *p* = 0.10, showing low heterogeneity among studies, so a fixed-effects model was applied for analysis. Meta-analysis showed that the combined effect size SMD = 0.22, 95%CI: (−0.08, 0.52), *p* = 0.15, and there was no significant difference between the two groups.

#### 3.4.5. Effects of Low Intensity Resistance Training with Blood Flow Restriction on Walking Ability in Middle-Aged and Older Adults

Four of the fourteen included papers (seven studies in total) compared the differences in group for low intensity resistance training with blood flow restriction with the control group for walking ability (Figure 7). The results of the heterogeneity test demonstrated that I^2^ = 85%, *p* < 0.10, indicating high heterogeneity among studies. Therefore, the random effects model was adopted for analysis. Meta-analysis revealed that the combined effect size SMD = −0.89, 95% CI: (−1.71, −0.06), *p* = 0.03, suggesting a statistically significant difference compared to the control group, and that low intensity resistance training with blood flow restriction was effective in improving walking ability in middle-aged and older adults.

To explore possible reasons for the heterogeneity, a subgroup analysis of walking ability was conducted. The 400 m walk test was used in two studies, the 6 m backward tandem walk test in two studies, and the TUG in three studies (Table 3). There was a significant difference between the experimental and control groups when only the TUG test method was adopted. There was no significant change in heterogeneity between the 400 m walk test and the TUG method, and the heterogeneity was significantly reduced when the 6 m backward tandem walk test method was adopted. It could be seen that the test method may be a source of heterogeneity and that the 6 m backward tandem walking capacity may be more sensitive to pressurization.

#### 3.4.6. Analysis of Subgroup Moderation Effects

Low intensity resistance training with blood flow restriction has positive effects on muscle strength, muscle mass of the lower extremities, and walking ability in middle-aged and older adults, which may be influenced by different age, exercise cycle, exercise frequency, exercise intensity, and vascular flow blocking pressure. Due to the lack of included studies on lower extremity muscle mass and walking ability and the similar characteristics of the moderating variables across studies, subgroups could not be set. Therefore, subgroup analysis of age, exercise cycle, exercise frequency, exercise intensity, and vascular flow resistance pressure for lower limb muscle strength index was performed in this paper.

As shown in Table 4, there were significant differences between the experimental and control groups at different exercise cycles and vascular flow blocking pressures. However, the differences were not statistically significant when the exercise frequency was 2 times/week and the exercise intensity was 35–45% 1RM. Low intensity resistance training with blood flow restriction at age 55–64 years, exercise cycle 4–8 weeks, exercise frequency 3 times/week, exercise intensity 20–30% 1RM, and vascular flow resistance pressure ≥ 120 mmHg was more likely to increase lower limb muscle strength.

#### 3.4.7. Publication of Bias Tests

A publication bias test was performed for lower limb muscle strength. Fewer than 10 studies were included for other outcome indicators, with insufficient test efficacy for publication bias testing. From the funnel plot (Figure 8), it can be seen that the graph is basically symmetrical. The result of Begg’s test showed that Z = 0.58, *p* > |z| = 0.559, *p* > 0.05, suggesting no publication bias in the study.

## 4. Discussion

In the middle-aged and older population, deterioration in the muscle strength, decreased muscle mass and functional performance of the lower extremities, impaired balance and slower walking speed are associated with the risk of falls. This study adopted meta-analysis to clarify the effect of low intensity resistance training with blood flow restriction on fall resistance in middle-aged and older adults based on the 14 included studies.

The results of the study showed that after LIRT + BFR intervention, muscle strength, muscle mass of the lower extremities and walking ability of the middle-aged and older adults were improved to different degrees, which was consistent with the previous results [11,14]. The possible effect mechanism of LIRT + BFR to improve muscle strength and mass is that the hypoxic environment during LIRT + BFR leads to increased metabolic stress, including the accumulation of lactic acid and reactive oxygen species, which promotes the secretion of growth hormone and other hormones related to muscle anabolism and activates mechanisms that induce muscle growth [31]. LIRT + BFR leads to an altered motor unit recruitment pattern during exercise, where the hypoxic muscle environment and high metabolite accumulation inhibit the recruitment of slow-twitch muscle fibers, which are predominantly aerobic, but mobilize fast-twitch muscle fibers to participate in muscle work [32]. The possible effect mechanism of LIRT + BFR to improve walking is that the increased muscle strength in the older adults improves the control of the joints of the lower limbs, which in turn maintains gait stability [33,34]. LIRT + BFR stimulates angiogenesis (increased serum concentrations of PECAM-1, CD34, VEGF-R2) and improves endothelial function (increased Flow-Mediated Dilatation (FMD)), thereby improving muscle function and walking distance [16,35]. 

The current study found that the intervention effect of LIRT + BFR on lower limb muscle strength was better than that of daily exercise and LIRT, but in comparison, the effect of HIRT was better than that of LIRT + BFR, which was generally consistent with the previous results [15]. HIRT promoted muscle strength increase mainly through high load stimulation to collect higher levels of high threshold motor unit involvement [36], which has been found that EMG amplitude was greater after HIRT than that after LIRT + BFR [37,38]. Fatela et al. [39] found that the median frequencies of rectus femoris after HIRT and LIRT + BFR respectively decreased by 12.5% and 10.5%, and the root mean square amplitude after HIRT was significantly higher than that after LIRT + BFR, so the muscle activation level of LIRT + BFR was lower than that of HIRT. HIRT maintained and increased muscle strength, but training at that intensity could easily cause conditions such as organism injury and intolerance in the older adults.

To further investigate the optimal exercise dose, the study analyzed the modulating effects of LIRT + BFR on lower limb muscle strength from the perspectives of exercise cycle, exercise frequency, exercise intensity and vascular flow blocking pressure. The intervention effect of low intensity resistance training with blood flow restriction on lower limb muscle strength was more significant when the exercise cycle was 4–8 weeks, exercise frequency was 3 times/week, exercise intensity was 20–30% 1RM, and vascular flow blocking pressure was ≥120 mmHg. In terms of exercise cycles, it usually took 10 weeks to achieve significant benefits for those with muscle hypertrophy when resistance training was performed. As exercise time increased, the body’s neuromuscular adaptations increased, muscle strength improved, and physical function enhanced [40]. Slysz et al. [12] also found that more than 6 weeks of BFR training produced greater strength gains. In terms of exercise frequency, resistance training was traditionally recommended two to four times per week to effectively induce skeletal muscle hypertrophy and strength increase. While for more than three weeks of resistance training with blood flow restriction, a training frequency of three times per week was sufficient to promote muscle hypertrophy effects and strength gains [40,41,42]. Exercise intensity determined the safety and effectiveness of resistance exercise prescriptions. Resistance training with blood flow restriction could promote muscle growth and improve muscle strength with loads of 20% to 40% of 1RM, while higher stresses might be required to induce muscle growth with the use of 20% of 1 RM [43]. In terms of vascular flow-blocking pressures, a large number of studies have used a range of 120 to 270 mmHg, with low-intensity resistance training being more effective at higher flow-limiting pressures.

Notably, there was a tendency to improve lower limb muscle function and balance after the LIRT + BFR intervention, but the effect was not significant. In terms of muscle function, Cook et al. [18] found the same results, while Zhang et al. [15] found that LIRT + BFR was effective in improving lower limb muscle function using meta-analysis. In terms of balance, Linero et al. [21] showed that 12 weeks of LIRT + BFR improved balance, and the effect was similar to that of moderate-to-high intensity resistance exercise. Yokokawa et al. [30] also found that 8 weeks of LIRT + BFR improved balance, and the effect was similar to that of dynamic balance exercise. On the one hand, the differences in the study results were influenced by the different control groups, and on the other hand it may be due to the fact that the increase in lower limb muscle strength after training in the subject population has not yet reached the threshold for improving muscle function and balance.

In conclusion, as a low-intensity and high-efficiency training modality, low intensity resistance training with blood flow restriction has selective positive effects on fall resistance in middle-aged and older adults, with significant advantages in improving lower limb muscle strength, lower limb muscle mass and walking ability. The following limitations exist in this study. (i) This study only included Chinese and English literature, which was slightly lacking in the comprehensiveness of the search. (ii) The study only included middle-aged and older adults over 50 years of age, with limited extrapolability of the findings. (iii) Besides lower limb muscle strength, fewer studies were included for other outcome indicators, so subgroup analysis and publication bias tests were not performed. Therefore, the effects of different cycles, frequencies, and intensities are not yet clear. (iv) The 14 studies included in the study were all small sample studies, with a lack of literature concerning large samples. There also existed a high degree of heterogeneity among studies, by which conclusions would be influenced to some extent.

## 5. Conclusions

LIRT + BFR can effectively improve lower limb muscle strength, muscle mass and walking ability, and can serve as an important form of anti-fall training for middle-aged and older adults. The intervention effect of LIRT + BFR on lower limb muscle strength was better than that of the daily exercise group and the low-intensity resistance exercise group, but the effect of high-intensity resistance exercise was superior compared to that of LIRT + BFR. In this paper, we investigated the dose-effect relationship between low intensity resistance training with blood flow restriction and lower limb muscle strength in terms of age, exercise cycle, exercise frequency, exercise intensity, and vascular blocking pressure, and found that the intervention effect of blood flow restriction intervention low-intensity resistance training on lower limb muscle strength was more significant at age 55–64 years, exercise cycle 4–8 weeks, exercise frequency 3 times/week, exercise intensity 20–30% 1 RM, and vascular blocking pressure ≥120 mmHg. Scientific exercise intervention is recommended for middle-aged and older adults as early as possible.

Although this study provides a basis for blood flow restriction intervention low-intensity resistance training in middle-aged and older adults, there is still a lack of standardized and more consistent schemes of blood flow restriction intervention low-intensity resistance training for middle-aged and older adults. In this paper, due to the small number of included literature, there still exists some uncertainty in the results of lower limb muscle function and balance, and more high-quality and large-sample randomized trials are needed to confirm them in the future.

## Figures and Tables

**Figure 1 ijerph-20-04723-f001:**
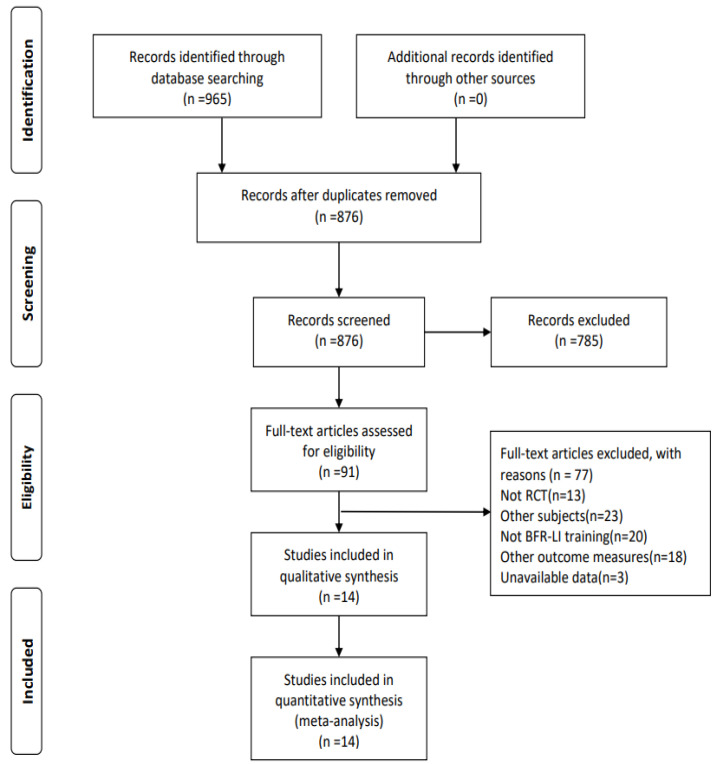
Flow chart of literature screening.

**Figure 2 ijerph-20-04723-f002:**
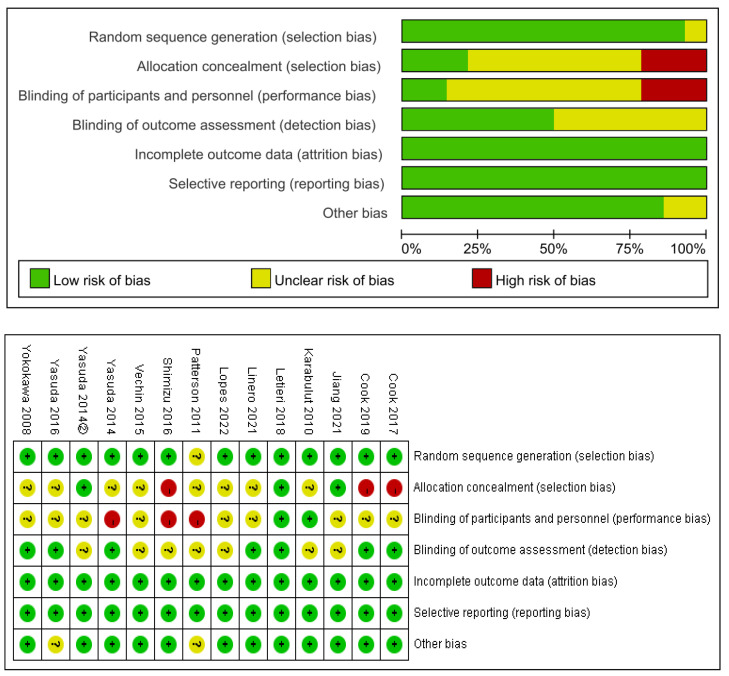
Risk of bias graph [10,18,19,20,21,22,23,24,25,26,27,28,29,30].

**Figure 3 ijerph-20-04723-f003:**
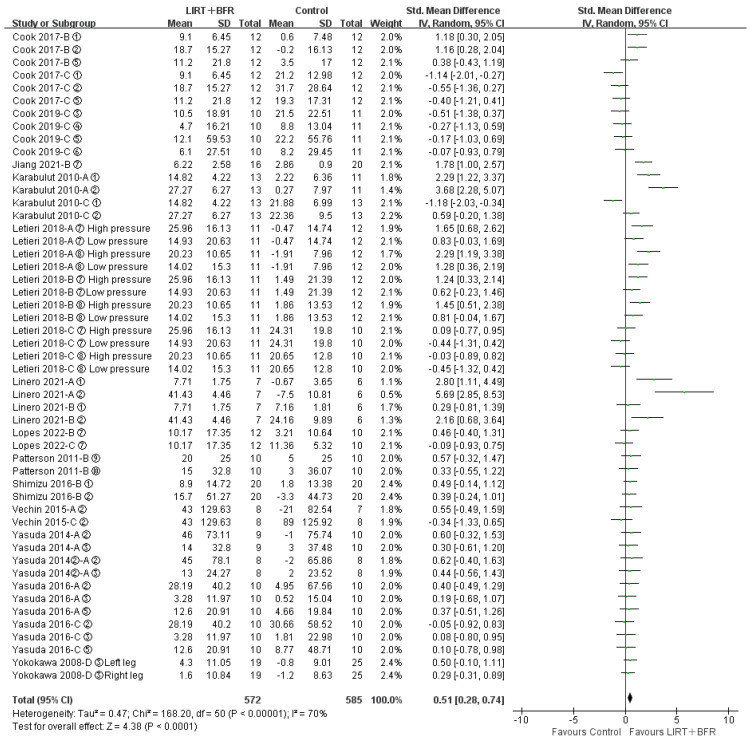
Meta-analysis of lower limb muscle strength [10,18,19,20,21,22,23,24,25,26,28,29,30]. Figure Notes: A, LIRT + BFR vs. daily exercise. B: LIRT + BFR vs. LIRT; C, LIRT + BFR vs. HIRT; D, LIRT + BFR vs. dynamic balance exercise. ① Leg extension-1RM (kg); ② Leg press-1RM (kg) ③ Knee extension-1/10RM (kg); ④ Knee flexion-1/10RM (kg); ⑤ knee extension-MVC (Nm); ⑥ Knee flexion-MVC (Nm); ⑦ Isokinetic knee extension (Nm); ⑧ Isokinetic knee flexion (Nm); ⑨ Dynamic plantar flexion-1RM (kg); ⑩ Dynamic plantar flexion-MVC (Nm).

**Figure 4 ijerph-20-04723-f004:**
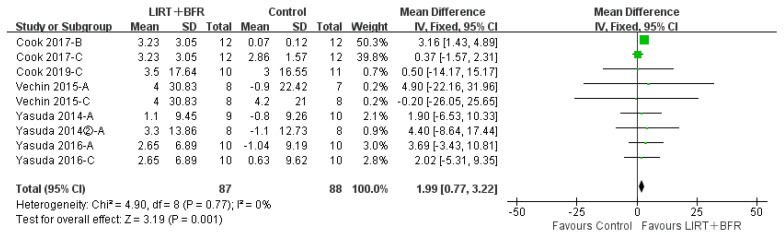
Meta-analysis of lower limb muscle mass [18,19,23,24,25,26]. Figure Notes: A, LIRT + BFR vs. Daily exercise; B, LIRT + BFR vs. LIRT; C, LIRT + BFR vs. HIRT. Quadriceps CSA (cm^2^).

**Figure 5 ijerph-20-04723-f005:**
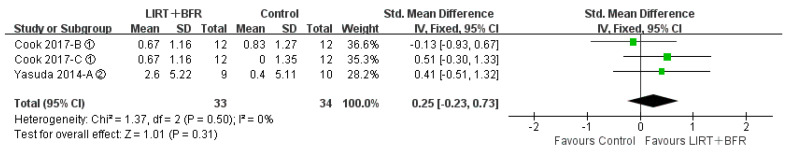
Meta-analysis of lower limb muscle function [18,24]. Figure Notes: A, LIRT + BFR vs. Daily exercise; B, LIRT + BFR vs. LIRT; C, LIRT + BFR vs. HIRT. ① SPPB; ② 30 s sit to stand test(s).

**Figure 6 ijerph-20-04723-f006:**
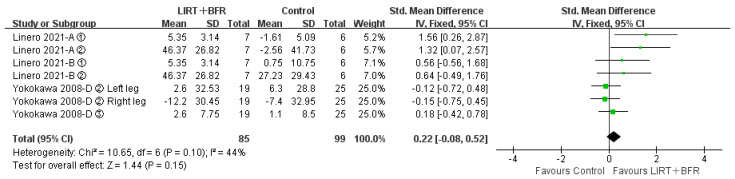
Meta-analysis of balance [21,30]. Figure Notes: A, LIRT + BFR vs. Daily exercise; B, LIRT + BFR vs. LIRT; D, LIRT + BFR vs. Dynamic Balance Exercise. ① Static eyes closed(s); ② Static eyes open(s); ③ Balance functional reach test.

**Figure 7 ijerph-20-04723-f007:**
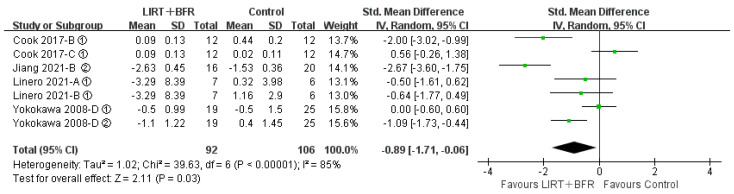
Meta-analysis of walking ability [18,21,27,30]. Figure Notes: A, LIRT + BFR vs. Daily exercise; B, LIRT + BFR vs. LIRT; C, LIRT + BFR vs. HIRT; D, LIRT + BFR vs. Dynamic Balance Exercise. ① Walking time(s). ② TUG(s).

**Figure 8 ijerph-20-04723-f008:**
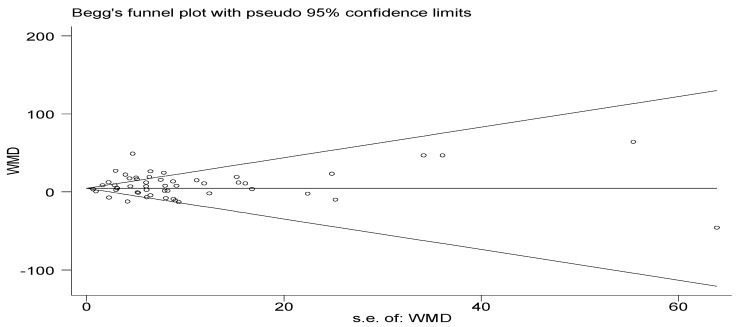
Begg’s test chart of lower limb muscle strength.

**Table 1 ijerph-20-04723-t001:** Basic information of the literature included in the meta-analysis.

Study	Country	Group	N	Age	Exercise Mode	%1RM	Duration (wk)	Frequency (days/wk)	Cuff Pressure (mmHg)	Measurements	
Cook et al. 2017 [18]	United States	LIRT + BFR	12	72.3–80.7	LE, LC, LP	LE:30 LC:30 LP:50	12	2	184 ± 25	Leg extension-1RM, Leg press-1RM, knee extension-MVC, Quadriceps CSA, SPPB, walking time	
		HIRT	12	71.3–82.0		70					
		Daily exercise	12	69.6–79.9							
Cook et al. 2019 [19]	United States	LIRT + BFR	10	76.4 ± 6.6	KE, KF	30	12	2	184 ± 25	Knee extension-10RM, Knee flexion-10RM, Knee extension-MVC, Knee flexion-MVC, Quadriceps CSA	
		HIRT	11	76.3 ± 8.7		70					
Karabulut et al. 2010 [20]	United States	LIRT + BFR	13	55.9 ± 1.0	LP, LE	20	6	3	205.4 ± 4.3	Leg press-1RM, Leg extension-1RM	
		HIRT	13	57.5 ± 0.8		80					
		Daily exercise	11	66 ± 5							
Linero et al. 2021 [21]	Korea	LIRT + BFR	7	55.71 ± 0.52	LP, LE	30	12	3	188 ± 9	Leg press-1RM, Leg extension-1RM, Static eyes open, Static eyes closed, walking time	
		LIRT	6	56.50 ± 0.99		30					
		Daily exercise	6	56.83 ± 0.70							
Patterson et al. 2011 [22]	UK	LIRT + BFR	10	67 ± 3	Dynamic plantar flexion	25	4	3	110	Dynamic plantar flexion-1RM, Dynamic plantar flexion-MVC	
		LIRT	10	67 ± 3		25					
Shimizu et al. 2016 [10]	Japan	LIRT + BFR	20	72 ± 4	LP, LE	20	4	3	134 ± 16	Leg press-1RM, Leg extension-1RM	
		LIRT	20	70 ± 4		20					
Vechin et al. 2015 [23]	Brazil	LIRT + BFR	8	65 ± 2	LP	20–30	12	2	71 ± 9	Leg press-1RM, Quadriceps CSA	
		HIRT	8	62 ± 3		70–80					
		Daily exercise	7	66 ± 5							
Yasuda et al. 2014 [24]	Japan	LIRT + BFR	9	71.3 ± 7.1	LP, KE	20–30	12	2	120–270	Leg press-1RM, Knee extension-1RM, Quadriceps CSA,30STS	
		Daily exercise	10	67.7 ± 6.0							
Yasuda et al. 2014② [25]	Japan	LIRT + BFR	8	70 ± 6	LP, KE	20–30	12	2	120–270	Leg press-1RM, Knee extension-1RM, Quadriceps CSA	
		Daily exercise	8	67 ± 7							
Yasuda et al. 2016 [26]	Japan	LIRT + BFR	10	70 ± 6	Squat, KE	35–45	12	2	161 ± 12	Leg press-1RM, Knee extension-1RM, Knee extension-MVC, Quadriceps CSA	
		HIRT	10	72 ± 7		70–90					
		Daily exercise	10	68 ± 6							
Jiang et al. 2021 [27]	China	LIRT + BFR	16	77.1 ± 6.60	KE, KF	30	8	3	80	Isokinetic knee extension, TUG	
		LIRT	20	76.45 ± 7.05		30					
Letieri et al. 2018 [28]	Brazil	LIRT + BFR	11	68.0 ± 3.79	Squat, LP, LC, KE	20–30	16	3	High pressure: 185.75 ± 5.45	Isokinetic knee extension, Isokinetic knee flexion,	
		LIRT + BFR	11	69.4 ± 5.73		20–30			Low pressure: 105.45 ± 6.5		
		HIRT	10	66.75 ± 4.43		70–80					
		LIRT	12	71.27 ± 4.73		20–30					
		Daily exercise	12	69.0 ± 6.39							
Lopes et al. 2022 [29]	Brazil	LIRT + BFR	12	71 ± 6	LP, KE	30	12	3	50% of Raop	Isokinetic knee extension	
		HIRT	10	73 ± 7		70					
		LIRT	10	72 ± 8		30					
Yokokawa et al. 2008 [30]	Japan	LIRT + BFR	19	72.3 ± 4.5	Half squat, KE, KF	20–25	8	2	70–150	Knee extension-1RM, Balance functional reach test, Static eyes open, TUG, walking time	
		Dynamic Balance Exercise	25	71.0 ± 4.1				1			

Abbreviations: LIRT + BFR, low-intensity resistance training with blood flow restriction; HIRT, high-intensity resistance training; LIRT, low-intensity resistance training; LE, leg extension; LC, leg curl; LP, Leg press; KE, knee extension; KF, knee flexion; wk, week/s; 1/10RM, 1/10-repetitive maximum strength; MVC, maximum voluntary contraction; CSA, cross-sectional area; SPPB, short physical performance battery; 30STS, 30s sit to stand test; TUG timed up and go test.

**Table 2 ijerph-20-04723-t002:** Effect of low intensity resistance training with blood flow restriction and different control groups on lower limb muscle strength.

Experimental vs. Control Group	N	SMD	95%CI	*p*	I^2^	Pheterogeneity
LIRT + BFR vs. Daily exercise	156/155	1.13	0.66, 1.60	<0.00001	70%	<0.00010
LIRT + BFR vs. LIRT	182/186	0.78	0.53, 1.03	<0.00001	24%	0.19000
LIRT + BFR vs. HIRT	196/194	−0.27	−0.47, −0.07	0.00900	0%	0.47000
LIRT + BFR vs. Dynamic Balance Exercise	38/50	0.39	−0.03, 0.82	0.07000	0%	0.62000

**Table 3 ijerph-20-04723-t003:** A subgroup analysis of the effects of low intensity resistance training with blood flow restriction on walking ability in middle-aged and older adults.

Test Method	N	SMD	95%CI	*p*	I^2^	PHeterogeneity
400 m walk test	24/24	−0.14	−0.55, 0.27	0.51	96%	<0.00001
6m backward tandem walk test	14/12	−4.05	−8.86, 0.76	0.10	0%	0.86000
TUG	54/70	−0.88	−1.61, −0.16	0.02	78%	0.01000

**Table 4 ijerph-20-04723-t004:** A subgroup analysis of the effects of low intensity resistance training with blood flow restriction on lower limb muscle strength in middle-aged and older adults.

Adjustment Variables	Subgroup Category	N	SMD	95%CI	*p*
Age	55–64 years old	80/72	1.82	0.52, 3.12	0.00600
	65–75 years old	492/513	0.38	0.18, 0.57	0.00010
exercise cycle	4–8 weeks	166/178	0.79	0.23, 1.35	0.00500
	12–16 weeks	406/407	0.43	0.18, 0.67	0.00060
exercise frequency	2 times/week	260/277	0.15	−0.06, 0.36	0.16000
	3 times/week	312/308	0.90	0.52, 1.27	<0.00001
exercise intensity	20–30%1RM	488/501	0.58	0.32, 0.84	<0.00010
	35–45%1RM	84/84	0.20	−0.14, 0.54	0.25000
vascular flow	<120 mmHg	142/143	0.47	0.09, 0.84	0.01000
resistance pressure	≥120 mmHg	430/444	0.56	0.27, 0.84	0.00010

## Data Availability

The data are from the published literature.

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
