# Peer review of "The Effect of Low Intensity Resistance Training with Blood Flow Restriction on Fall Resistance in Middle-Aged and Older Adults: A Meta-Analysis"

_ijerph, 2023, doi:10.3390/ijerph20064723_

Round 1
Reviewer 1 Report
The study apparently conducted the systematic review with care. They used several bases, the PICOT structure, and presented the search system in detail. In a quick analysis, I did not find recent meta-analyses with the same objective, increasing the study's innovation. However, the text is difficult to follow, with long, wordy sentences. The entire article needs revision in order to reduce sentences and make the text more fluid. Not least, the figures and tables are presented with poor quality. It is highly recommended that this be adjusted.
Line 55 – Strong citations are required here, especially when using the term high safety.
Line 55 – Numerous? Only three were cited.
Line 69 – “…at home and abroad…”?
Line 67 – 74 – These sentences are unnecessary.
1.2.1 – Do not use the subsequent structure. Based on the topics I understood that the PICOT structure was used. Re-arrange the paragraph explaining this.
Line 113 - And if a particular study did not conduct an exercise with the control group, would that study be included?
Line 126 – “non-Chinese and English literature”. I honestly do not understand what this means. Were only studies in English included?
Line 126 – 127 – “duplicate published literature”. Same thing here.
Line 127 – 128 – Same for item 5.
Line 133 – Which authors? Provide the initials. Same thing on line 135.
Lines 168-175 – Adjust this entire paragraph. It is hard to follow.
Table 1 – Improve the quality of table 1, including the name of variables. What â‘¡ mean?
Figure 3 - Maybe I missed something, but why does the same article appear multiple times in meta-analyses with different input data? This needs to be made clear.
Author Response
According to the revision opinions proposed by the expert, our team carefully studied and discussed, and revised the raised issues one by one. We appreciate the pertinent comments from expert, which benefited our team a lot.
Point 1: The study apparently conducted the systematic review with care. They used several bases, the PICOT structure, and presented the search system in detail. In a quick analysis, I did not find recent meta-analyses with the same objective, increasing the study's innovation. However, the text is difficult to follow, with long, wordy sentences. The entire article needs revision in order to reduce sentences and make the text more fluid. Not least, the figures and tables are presented with poor quality. It is highly recommended that this be adjusted.
Response 1: We appreciate the advice of the experts, and we have revised and streamlined the article, enlisting the assistance of English language experts for extensive editing of the English language and style, and standardizing the figures and tables.
Point 2: Line 55–Strong citations are required here, especially when using the term high safety.
Response 2: We thank the experts for their suggestions, and we will use the wording with care and add references. Low intensity resistance training with blood flow restriction (LIRT + BFR) has the characteristics of low intensity, high efficiency and low mechanical stress on joints and cardiovascular system, which is relatively safe compared to traditional high-intensity resistance training (Reference: 10.1007/s00421-016-3328-8).
Point 3: Line 55–Numerous? Only three were cited.
Response 3: We would like to thank the experts for their comments, and we will be careful with the wording and revise the phrase "a large number of studies" to "existing studies".
Point 4: Line 69 – “…at home and abroad…”?
Response 4: We appreciate the expert's comments. A clerical error was made in the article and has been removed.
Point 5: Line 67 – 74 – These sentences are unnecessary.
Response 5: Thanks for the expert advice, it has been removed.
Point 6: 1.2.1 – Do not use the subsequent structure. Based on the topics I understood that the PICOT structure was used. Re-arrange the paragraph explaining this.
Response 6: Thanks to the expert's suggestion, the inclusion criteria of the literature are based on PICOS principles and have been revised and marked in the article.
Point 7: Line 113 - And if a particular study did not conduct an exercise with the control group, would that study be included?
Response 7: Thanks to the expert for the question, the text does lack clarity in its presentation. If a study control group does not perform exercises to maintain activities of daily living, and that control group produces physical activity or exercise behaviors while maintaining daily living, then that control group is included and included in the daily exercise group.
Point 8: Line 126 – “non-Chinese and English literature”. I honestly do not understand what this means. Were only studies in English included?
Response 8: Thanks to the experts' questions, the included studies include English and Chinese literature and do not include literature in other languages.
Point 9: Line 126 – 127 – “duplicate published literature”. Same thing here.
Response 9: Appreciate the expert's question. Duplicate publication refers to a paper in which an author publishes a paper that is substantially similar to his or her own published paper, and without attribution or consent from the original copyright holder. There may be meaningless differences between the original paper and the new paper, for example, by using a new title or a modified abstract, but the data set and findings are identical. Duplicate publications were not included.
Point 10: Line 127 – 128 – Same for item 5.
Response 10: We appreciate the expert's question. Item 5, "The experimental data are incomplete and cannot be calculated" means that the description of the experimental data is not clear and the experimental data cannot be extracted for analysis, so this kind of literature is not included.
Point 11: Line 133 – Which authors? Provide the initials. Same thing on line 135.
Response 11: Thanks for the expert advice. Authors' initials have been added and marked in the article. The literature was screened by 2 researchers (XX, XJZ) according to the inclusion and exclusion criteria, respectively, and the information was extracted and cross-checked. In case of disagreement, a third person (JW) jointly discussed the inclusion decision.
Point 12: Lines 168-175 – Adjust this entire paragraph. It is hard to follow.
Response 12: Thanks for the expert advice. The text is indeed not clear enough, and we have revised and marked it.
Point 13: Table 1 – Improve the quality of table 1, including the name of variables. What â‘¡ mean?
Response 13: Thanks for the expert advice. We recreated this table and wrote the names of the variables clearly. Two papers published by the author Yasuda in 2014 were included in this study, and the second one was labeled as â‘¡ in order to distinguish the literature.
Point 14: Figure 3 - Maybe I missed something, but why does the same article appear multiple times in meta-analyses with different input data? This needs to be made clear.
Response 14: Thanks to the expert for the question. There are parts of the literature where there are multiple intervention groups and we have analyzed the effect of LIRT+BFR compared to different control groups and multiple test metrics represent the same outcome metric, therefore this occurs. We have further normalized the figure.

Reviewer 2 Report
Thank you for your submission.
This manuscript reports a meta-analysis of 14 RCTs related to low-intensity BFRT as an anti-fall intervention in middle-aged and older adults. The results show that LIRT+BFR can be used as an important form of anti-fall training. This manuscript is incredibly clear and concise, and the authors show a revealing command of previously published studies. The authors have done a great job of providing an informative and meaningful addition to the current study field.
I only have a couple of questions that may help elevate the overall contribution of the paper to this research field.
General questions:
Is the title appropriate? BFR on anti-fall intervention? It sounds a little strange to me.
Is it appropriate to term adults over or equal to 50 years old "middle-aged"?
Line 303 - lactate? The body actually produces lactate, which is lactic acid, minus one proton.
Line 346 & 353 - [42] & [20]- superscript?
Lastly, I would reread everything for spelling and punctuation.
Very nice work.
Author Response
According to the revision opinions proposed by the expert, our team carefully studied and discussed, and revised the raised issues one by one. We appreciate the pertinent comments from expert, which benefited our team a lot.
Point 1: Is the title appropriate? BFR on anti-fall intervention? It sounds a little strange to me.
Response 1: Thanks to the expert's suggestion, the title has been changed to " The Effect of Low Intensity Resistance Training with Blood Flow Restriction on Fall Resistance in Middle-Aged and Older Adults: A Meta-Analysis".
Point 2: Is the title appropriate? BFR on anti-fall intervention? It sounds a little strange to me.
Response 2: We thank the experts for their proposal. The setting of 50 years is based on the support of the literature (Reference:10.1161/CIRCULATIONAHA.107.185650). In accordance with the World Health Organization's classification criteria for each age group (45-59 years as middle-aged, 60-74 years as younger older adults, 75-89 years as general older adults, and 90 years or older as long-lived older adults), we added specific age groups for middle-aged and older adults in the inclusion criteria section.
Point 3: Line 303 - lactate? The body actually produces lactate, which is lactic acid, minus one proton.
Response 3: Thanks for the expert advice. It refers to lactic acid. LIRT+BFR causes local hypoxia in tissues, which in turn promotes a rise in the concentration of lactic acid in blood and muscle in the body.
Point 4: Line 346 & 353 - [42] & [20]- superscript?
Response 4: Thank you for the expert advice. It was an oversight, and we have superscripted and marked it in the text.

Reviewer 3 Report
Dear authors,
The submited paper raises important issue of fall raisk in middle aged and senior population and potential role of BFR training that could play in porces of rehabilitation and fall prevention practise.
In general the submited manuscript is based aon well-designed methodology and well written, but some changed and improvements are required.
1)Line 49 - BFR is not exactly the same as Kaatsu training. In original Kaatsu techniqe the pressure used during training is higher than SBP and causes occlusion or significant impairmant in blood inflow. Modern BFR is using lower pressure only to obstruct the venous outflow.
2) Line 55-60: BFR is also known to influence the vascular functions, especially the endothelial parameters and angiogenesis. Those are relevant information as improved vascular parameters are beneficial for muscles' functions. I would suggest to add reference i.ex.: 10.3390/ijerph192315859
3) Line 109: Why was it 50 years old? Was it arbitraty set or based on literature?
4) Line 111: How were the comorbidities in all searched articles checked?
5) Figures 3-7 are unclear. The same studies are used multiple times (probably due to different populations/training methods used in those studies) but they are not sufficiently described. The reader is unable to identify which line represents which population - it needs to be clarified and improved. Also units are missing.
6) Line 301-311 - this sentence is way to long and unreadable. It needs to be splited. Also information on impact of BFR on angiogensis factors is missing (I suggest to re-use reference: https://doi.org/10.3390%2Fijerph192315859) Also influence of BFR on endogelin production is worth adding as it is known to improve the walking distance (reference: 10.1016/j.jash.2017.06.009)
Your sincerely
Author Response
According to the revision opinions proposed by the expert, our team carefully studied and discussed, and revised the raised issues one by one. We appreciate the pertinent comments from expert, which benefited our team a lot.
Point 1: Line 49 - BFR is not exactly the same as Kaatsu training. In original Kaatsu techniqe the pressure used during training is higher than SBP and causes occlusion or significant impairmant in blood inflow. Modern BFR is using lower pressure only to obstruct the venous outflow.
Response 1: Thanks for the expert advice. It is true that BFR originated from KAATSU and is not exactly the same as Kaatsu training, as the expert said. We have revised and marked in the text.
Point 2: Line 55-60: BFR is also known to influence the vascular functions, especially the endothelial parameters and angiogenesis. Those are relevant information as improved vascular parameters are beneficial for muscles' functions. I would suggest to add reference i.ex.: 10.3390/ijerph192315859.
Response 2: We thank the experts for their suggestions. Our team studied the references carefully and made the appropriate changes in the text. It is true that LIRT + BFR can stimulate angiogenesis and improve endothelial function, which benefits muscle function, as the experts suggest. We will focus on this in future studies.
Point 3: Line 109: Why was it 50 years old? Was it arbitraty set or based on literature?
Response 3: We thank the experts for their proposal, and the setting of 50 years is based on the support of the literature (Reference:10.1161/CIRCULATIONAHA.107.185650). According to the World Health Organization's classification criteria for each age group (45-59 years as middle-aged, 60-74 years as younger older adults, 75-89 years as general older adults, and 90 years or older as long-lived older adults), we added specific age groups of middle-aged and older adults in the inclusion criteria section. We performed a subgroup analysis of age groups to explore the effects of LIRT + BFR on middle-aged and older adults at different ages, and the results showed that low intensity resistance training with blood flow restriction at ages 55-64 years better increased lower extremity muscle strength. The earlier the exercise intervention was performed in middle-aged and older adults, the better the results were.
Point 4: Line 111: How were the comorbidities in all searched articles checked?
Response 4: We thank the experts for their advice. The subjects of the 14 studies included in the analysis were healthy older adults, free of psychiatric abnormalities and severe perceptual disturbances, without musculoskeletal disorders and surgical history, and without major organic diseases. The comorbidities in the examination articles were mainly described by careful reading of the participants in the methods.
Point 5: Figures 3-7 are unclear. The same studies are used multiple times (probably due to different populations/training methods used in those studies) but they are not sufficiently described. The reader is unable to identify which line represents which population - it needs to be clarified and improved. Also units are missing.
Response 5: Thanks to the expert for the question. There are parts of the literature where there are multiple intervention groups and we analyzed the effect of LIRT + BFR compared to different control groups and multiple test metrics represent the same outcome metric, hence this situation. In addition, we further described the basic information of the included studies and added units.
Point 6: Line 301-311 - this sentence is way to long and unreadable. It needs to be splited. Also information on impact of BFR on angiogensis factors is missing (I suggest to re-use reference: https://doi.org/10.3390%2Fijerph192315859) Also influence of BFR on endogelin production is worth adding as it is known to improve the walking distance (reference: 10.1016/j.jash.2017.06.009).
Response 6: Thanks to expert advice. We have streamlined and divided this paragraph. Our team carefully studied the references you provided, and this is a good addition. It is true that LIRT + BFR stimulates angiogenesis (increased serum concentrations of PECAM-1, CD34, VEGF-R2) and improves endothelial function (increased Flow-Mediated Dilatation (FMD)), thus improving muscle function and walking distance, as stated by the expert. (reference: https://doi.org/10.3390%2Fijerph192315859) (reference: 10.1016/j.jash.2017.06.009)

Round 2
Reviewer 1 Report
---The authors answered all my questions, as well as made changes to the study. Some important questions still persist, and it is imperative that these are improved in studies.
- Line 51-57 – These sentences are still wordy. Try to break or synthesize them.
-Line 131 - This is not yet clear to me. We usually only adopt the English language in systematic revisions. This standardizes the information, establishing a clear criterion for the language. When the authors also incorporate the Chinese language, they open the margin for severe criticism. Because they were in the Chinese language, only the Chinese managed to evaluate the studies included if they wish to read these. I can therefore question why, why Italian or German articles were not included, for example. This is a considerable problem and needs to be explained in the study.
-Authors answered that “Duplicate publication refers to a paper in which an author publishes a paper that is substantially similar to his or her own published paper, and without attribution or consent from the original copyright holder. There may be meaningless differences between the original paper and the new paper, for example, by using a new title or a modified abstract, but the data set and findings are identical. Duplicate publications were not included.” I am fully aware of duplicates, triplicates, and so on. The problem resides in the ambiguous nature of the words used. Have you excluded duplicate studies? The answer is no. Duplicate analysis, triplicates (etc ..) is part of the screening, not a criterion for exclusion. These are distinct things.
-Same thing for item 5. You need to explain the criteria, not merely present them.
Author Response
Response to Reviewer 1 Comments
According to the revision opinions proposed by the expert, our team carefully studied, discussed, and revised the raised issues one by one. We appreciate the pertinent comments from the expert, which benefited our team a lot.
Point 1: - Line 51-57 – These sentences are still wordy. Try to break or synthesize them.
Response 1: Thanks to expert advice, we have streamlined and revised the sentences and marked them in the manuscript.
Point 2: -Line 131 - This is not yet clear to me. We usually only adopt the English language in systematic revisions. This standardizes the information, establishing a clear criterion for the language. When the authors also incorporate the Chinese language, they open the margin for severe criticism. Because they were in the Chinese language, only the Chinese managed to evaluate the studies included if they wish to read these. I can therefore question why, why Italian or German articles were not included, for example. This is a considerable problem and needs to be explained in the study.
Response 2: We thank the expert for their suggestion. We added to the limitations of the manuscript: this study included literature in English and Chinese, literature in other languages was not included, and the comprehensiveness of the search was somewhat lacking. Our team will be more rigorous in future studies.
Point 3: -Authors answered that “Duplicate publication refers to a paper in which an author publishes a paper that is substantially similar to his or her own published paper, and without attribution or consent from the original copyright holder. There may be meaningless differences between the original paper and the new paper, for example, by using a new title or a modified abstract, but the data set and findings are identical. Duplicate publications were not included.” I am fully aware of duplicates, triplicates, and so on. The problem resides in the ambiguous nature of the words used. Have you excluded duplicate studies? The answer is no. Duplicate analysis, triplicates (etc ..) is part of the screening, not a criterion for exclusion. These are distinct things.
Response 3: We thank the expert for their suggestion, and the presentation in the manuscript is indeed not clear enough. We excluded duplicate publications from the literature screening phase. As the expert said, this is part of the literature screening, not an exclusion criterion. We have put the relevant statements in the literature screening.
Point 4: -Same thing for item 5. You need to explain the criteria, not merely present them.
Response 4: We thank the expert for their suggestion. Item 5, "Incomplete experimental data, unable to calculate", refers to the missing data of the mean, standard deviation, and sample size of the outcome indicators before and after the intervention in the literature when extracting the data, and the original data were not available after contacting the authors. We have explained and modified the exclusion criteria.
Reviewer 3 Report
Dear authors,
Thank you for the improvements that you introduced. At present form, the study is more precise and understandable for the readers.
I have no further requests or suggestions.
Congratulations on your work!
Yours sincerely
Author Response
Dear Expert,
We appreciate the pertinent comments from expert, which benefited our team a lot.
Best regards!